# Scoping Review of Climate Change Adaptation Interventions for Health: Implications for Policy and Practice

**DOI:** 10.3390/ijerph21121565

**Published:** 2024-11-26

**Authors:** Nicholas Brink, Kehkashan Mansoor, Joost Swiers, Darshnika P. Lakhoo, Craig Parker, Britt Nakstad, Shobna Sawry, Kristin Aunan, Ilona M. Otto, Matthew F. Chersich

**Affiliations:** 1Wits Planetary Health Research, University of the Witwatersrand, Johannesburg 2193, South Africa; 2Wits RHI, University of the Witwatersrand, Johannesburg 2193, South Africa; 3Wegener Center for Climate and Global Change, University of Graz, 8010 Graz, Austria; kehkashan.mansoor@edu.uni-graz.at (K.M.);; 4&flux B.V., 3011 KD Rotterdam, The Netherlands; 5Paediatrics and Adolescent Health, University of Botswana, Gabarone UB 0022, Botswana; 6Division of Pediatric and Adolescent Medicine, University of Oslo, 0316 Oslo, Norway; 7Centre for International Climate and Environmental Research, 0316 Oslo, Norway; 8Public Health and Primary Care, School of Medicine, Trinity College Dublin, D02 PN40 Dublin, Ireland

**Keywords:** climate change, adaptation interventions, health policy, scoping review

## Abstract

Climate change is among the greatest threats to health in the 21st century, requiring the urgent scaling-up of adaptation interventions. We aim to summarise adaptation interventions that were funded by the Belmont Forum and the European Union, the largest global funders of climate change and health research. A systematic search was conducted (updated February 2023) to identify articles on adaptation interventions for health within this funding network. The data extracted included study characteristics, types of interventions, and study outcomes. The results were synthesised narratively within the PRISMA-ScR guidelines. A total of 197 articles were screened, with 37 reporting on adaptation interventions. The majority of interventions focused on the general population (n = 17), with few studies examining high-risk populations such as pregnant women and children (n = 4) or migrants (n = 0). Targeted interventions were mostly aimed at behavioural change (n = 8) and health system strengthening (n = 6), while interventions with mitigation co-benefits such as nature-based solutions (n = 1) or the built environment (n = 0) were limited. The most studied climate change hazard was extreme heat (n = 26). Several studies reported promising findings, principally regarding interventions to counter heat impacts on workers and pregnant women and improving risk awareness in communities. These findings provide a platform on which to expand research and public health interventions for safeguarding public health from the effects of climate change.

## 1. Introduction

Climate change is increasingly recognised as the largest threat to the natural environment and human health in the 21st century. The Intergovernmental Panel on Climate Change provides incontrovertible evidence on the contribution of anthropogenic activities to climate change and provides insights into its wide-ranging effects [1]. Climate change not only contributes to incremental changes in global temperatures and rising sea-levels, but it also increases the frequency and intensity of extreme weather events, resulting in significant changes in disease patterns and negative impacts on essential resources such as food and water.

The Lancet Countdown provides a comprehensive overview of the current and future effects of climate change on health and associated outcomes [2]. Their findings indicate an 86% increase in heat-related mortality among the elderly since the baseline period of 1990–2000. Recent heatwaves in Europe are of particular concern; during the summer of 2022 alone, there were 61,672 heat-related deaths across Europe [3]. In addition, climate change increases the transmission risk for many infectious diseases, such as malaria, dengue, and vibriosis. The Countdown noted an increased potential for the transmission of dengue by up to 37% by 2050 [2]. Food security is also of profound importance, with the report noting an increase of 127 million people who experienced moderate-to-severe food insecurity in 2021 compared to the previous 40-year period.

Given the limited adoption of mitigation strategies, adaptation interventions are crucial for minimising the health impacts of climate change and enhancing resilience within communities. However, investment in adaptation measures with health benefits remains limited, with only USD 32 million of the Green Climate Fund used for health-related projects and only 6% of overall adaptation financing mechanisms [4]. Understanding the effectiveness of adaptation interventions is vital for informing evidence-based policy and practice. The evidence base, however, remains limited and often fragmented, potentially contributing to a lack of investment. A recent scoping review identified only 33 articles on adaptation interventions for health, with significant gaps in climate-sensitive infectious diseases and in the Global South [5].

The Belmont Forum and European Union represent the largest global funders of climate change and health research, and they have had a significant role in shaping the research agenda. Projects funded by these organisations have made great strides in understanding the impacts of climate change on health and adaptation interventions across a range of settings and topics. They have coordinated their efforts through programmes like the ENBEL project (ENhancing BELmont Research Action to support EU policymaking regarding climate change and health) [6]. Leveraging this research network to catalyse investment in climate–health financing requires an understanding of the current priorities and state of the field.

We aim to review the literature on adaptation interventions within this funding network to establish the types of interventions implemented, the target populations, the measured outcomes, and the implications for policy and practice. This will serve to guide future research funding and research agendas in the climate and health nexus.

## 2. Materials and Methods

A literature review was conducted following the principles in the Preferred Reporting Items for Systematic Reviews extension for Scoping Reviews (PRISMA-ScR) guidelines [7]. A comprehensive search strategy was developed in conjunction with the funding bodies to identify relevant projects within the Belmont Forum, EU Horizon 2020, and ERA4CS programmes, and was updated until February 2023. Thereafter, the principal investigators of the respective projects were contacted to provide articles related to the project. Any missing information that was noted during data extraction was requested from the principal investigators through electronic correspondence.

Two independent reviewers screened and identified articles for inclusion. The inclusion criteria encompassed articles that focused on adaptation interventions related to climate change and health and reported primary research or intervention evaluations. We included systematic and other review articles that summed existing evidence on an intervention within an analytic framework. No language restrictions were applied.

Data were extracted from the selected articles using a standardised form (Appendix A). The extracted information included study characteristics, types of interventions, and study outcomes. The study characteristics included project name, article title, location, population group, temporal focus (date or season of intervention, if relevant), research problem or main topic covered, and study objective. The intervention data extracted included a description of the intervention, the evaluation methodologies, and the main results of the intervention, including any negative or unintended findings.

The extracted data were analysed thematically, focusing on the types of adaptation interventions, populations studied, outcomes assessed, and implications for policy and practice. Interventions were classified by intervention type (health systems and new health services, built environment, poverty reduction and equity, nature-based solutions, policy actions, and behaviour change and awareness) and level of action (individual and household, facility, community and workplace, district and province, and national and international). Figure 1 below highlights these two components, where each report would be classified according to one or more components and level of action: for example, a nature-based solution at the community level.

The findings were synthesised according to the identified themes, and relevant quantitative data were summarised descriptively. As this study involved a review of published articles, no ethical approval was required.

Regarding the use of artificial intelligence, ChatGPT [8] (Large Language Model) was used to compile an initial draft by summarising the contents of a lengthier report on which this article is based. All outputs, for which we take full responsibility, have been reviewed and edited by the authors.

## 3. Results

This review includes articles published between October 2021 and February 2023, covering the Belmont Forum (n = 10), Horizon2020 (n = 4), and ERAC4S (n = 1) projects. A total of 194 published journal articles were identified for screening, of which 78 articles (40%) reported on adaptation interventions and were included for full-text review (Figure 2). However, 41 were subsequently excluded as they did not assess health adaptations. A total of 37 (19%) articles were included in this report. Of these, 20 articles reviewed potential interventions, and 17 articles reported directly on evidence for interventions implemented.

### 3.1. Timelines

The majority of articles were published between 2020 and 2021 (n = 18). Between 2020 and 2023, there were approximately an equal number of papers on implemented interventions and intervention reviews. However, interestingly, reports of implemented interventions dominated from 2016 to 2019, possibly due to the different focus areas of earlier funding calls.

### 3.2. Geographic Distribution

Most of the studies were conducted in the Global North (Figure 3). All studies conducted in Asia and North America were on implemented interventions, compared to 60% of studies in Europe. In Africa, by contrast, the studies were primarily intervention reviews.

### 3.3. Methodologies

The most common study methods utilised in the reports were reviews and intervention evaluations (n = 28), followed by surveys and interviews (n = 8) and modelling analyses (n = 6), with some studies employing mixed methodologies. One report was categorised as a cost-effectiveness study. Only 15 of 37 articles included a sub-group analysis or an inequity assessment.

### 3.4. Population Groups

The primary population group examined was the general population (n = 17). Among the high-risk groups studied, notable attention was given to occupational settings (n = 13). Other high-risk groups included urban residents, pregnant women, and children, although research in these groups constituted less than 20% of the articles. No reports examined the migrant population.

### 3.5. Exposure

The majority of reports focused on heat exposure (n = 26), while 11 reports focused on other extreme conditions or weather events, with a similarly equal proportion of implemented and reviewed interventions.

### 3.6. Topic of Study

The most common topic studied was adaptation interventions to reduce the impact of heat exposure (n = 15), followed by studies examining interventions to raise risk awareness (n = 14). Warning systems were predominantly covered in intervention reviews (n = 5). On the other hand, decision-making tools (n = 2) and disaster reduction interventions (n = 2) were primarily assessed in intervention studies using empirical data.

### 3.7. Intervention Framework

We categorised the thirty-seven adaptation studies by the six intervention components and six levels of action described in Appendix A. Some articles had multiple intervention components and levels of action. Through a thematic review, the specific intervention component breakdown was as follows: behaviour change and awareness (n = 8), health systems and new health services (n = 5), poverty reduction and equity (n = 5), and nature-based solutions (n = 1). No articles reported on an intervention component within the built environment.

Most interventions took place at the level of the community and workplace (n = 10). Four interventions were carried out at the facility level, three at the individual or household level, and one at the district or provincial level. No interventions were carried out at the national or international level.

Figure 4 provides a visual summary of the articles’ characteristics based on population group and adaptation topic. The left pie-charts show the relative number for each topic, and the right ones show the relative number for each population group, additionally split by implemented interventions versus intervention reviews. Additional figures, including comprehensive bar-charts on project name, year of publication, type of exposure, and methodology, are available in Appendix A.

### 3.8. Project Summaries

The HEAT-SHIELD project contributed the most studies (n = 15), while the Micro-Poll and CCEHN projects each contributed the least to this review, with only one study each. All the studies in the BuildERs, CCEHN, ClimAPP, Micro-Poll, and PREP presented findings of implemented interventions. However, the CHAMNHA and S&CC projects, by contrast, solely included intervention reviews or frameworks. The remaining projects had approximately an even number of reports on implemented interventions and reviews

Table 1 summarises the adaptation interventions studied, grouped by project. The BuildERS project provided evidence to support the use of novel technologies in disaster response efforts, where artificial intelligence and mobile technology showed significant benefits despite potential limitations of their use [9,10]. The CCEHN project examined the development of Early-Warning Systems and concluded that an innovative, multisectoral, local system approach to co-production can foster long-term engagement and nurture a culture of preparedness to achieve real risk reduction [11]. The CHAMNHA project provided insights into adaptation interventions targeted at high-risk populations: pregnant women, newborns, and children. In support of the findings of CCEHN, they emphasised the benefits of co-production and interventions suitable for the local context, where low-cost health education was at the forefront of the studied interventions [12,13,14,15]. ClimApp presents a novel heat early-warning system that provides a personalised risk profile and targeted interventions through a smartphone application, which was well received by testing groups [16,17]. This supports the findings of the BuildERS project on the benefit of novel technologies in adaptation interventions. The EXHAUSTION project focused on air conditioning as an intervention for the prevention of heat- and air pollution-related disease, confirming its effectiveness in reducing mortality [18]. Models were also evaluated to provide real-time disaster response parameters in air pollution linked to wildfires [19] with good correlation between the predicted and actual measurements. HEAT-SHIELD contributed significantly to the literature on early-warning systems and occupational heat risks and interventions [20,21,22,23,24,25,26,27,28,29,30,31,32,33,34]: for example, they provided an assessment of the effects of work capacity on rising heat indices, finding that the Universal Thermal Climate Index (UTCI) and the Wet Globe Bulb Temperature (WGBT) were best able to predict work capacity in outdoor environments, with an exponential decrease in work capacity with rising temperatures [35]. The S&CC project assessed the effect of climate change on schistosomiasis and found multiple effective interventions to reduce the burden of disease through a one-health approach. Interventions ranged from AI-assisted field identification of the helminth to the modification of the host environment through invasive species reduction and the introduction of biological predators, as well as community-based awareness and mass drug administration campaigns [36,37,38,39,40,41]. The PREP project found simple rest, shade, and water provision interventions most effective in reducing heat exposure in sugarcane field workers—a population with a disproportionately high burden of chronic kidney disease of unknown origin, although undertaking manual labour in hot conditions is likely a significant contributing factor [42,43,44,45]. Importantly, this study noted increased productivity with their interventions, highlighting important co-benefits. Micro-Poll examined the importance of pollinators in food security in rural populations and emphasised the importance of ecosystem management in overall adaptation to climate change for smallholder farmers [46].

## 4. Discussion

The overall body of work examined in this review is substantial in its scope, although limited in its depth. This review highlights multiple promising interventions that can be implemented on national scales, across multiple different exposure-outcome pathways. However, research on adaptation interventions constituted a minority of the literature published by EU- and Belmont-funded projects, highlighting a gap in current research priorities within the field.

### 4.1. Strengths and Weaknesses of Adaptation Interventions

Geographically, the studies cover a wide range of population groups from Asia, Africa, America, and Europe; however, the Global South remains relatively under-represented. Cross-comparisons between different geographic and socioeconomic settings would provide valuable insights, but the included studies seldom covered multiple centres and diverse climate hazards. However, the research sites involved in the projects in Africa, Asia, and South America may serve as platforms for future collaborative research in these under-represented regions and reduce bias towards the Global North [47].

The research projects reviewed drew upon a broad range of disciplines, including public health, social sciences, natural sciences, data science, and computer science, with increasing representation of artificial intelligence techniques—a component proving to be increasingly important in this field. Health economics methods were applied in only one project [23] to assess the relative costs and benefits of interventions, highlighting the need for future research in this field. An understanding of economic feasibility will be critical in financing future intervention packages. In most studies, mixed-method research was central to the methodology, providing a more nuanced understanding of the focus area. This underscores the importance of close interdisciplinary collaboration.

The interventions focused largely on behaviour change and health system strengthening, mainly targeting the community or workplace. This focus neglected important interventions in nature-based solutions and adaptations to the built environment, which could be an important priority in addressing climate change through combined adaptation–mitigation interventions. Despite this, the MicroPoll and S&CC projects showed promising results in assessing a package of interventions aiming to adapt the natural environment to improve human health. In addition, the BuildERS, EXHAUSTION, and HEAT-SHIELD projects focus on increasing societal resilience by strengthening the social capital, risk awareness, and preparedness of vulnerable segments of society. This highlights the importance of embracing many diverse intervention options in the climate and health nexus. The wide range of exposures and settings across studies within this review has limited overall coherence. The lack of focused attention may hinder researchers, funding agencies, and policymakers to discern priorities. However, given the field’s ongoing development, the projects will prove to be an essential step in identifying such priorities and providing a base to launch future research.

Extreme heat was the most common climate change hazard addressed by the projects, illustrating this exposure as an important priority of the field, but potentially at the detriment of other important exposures such as air pollution and extreme weather events. Despite the focus on the general population, the findings provide a broader understanding of which sub-groups are more vulnerable to climate change’s impacts, such as pregnant women, children, and occupational workers. This will prove helpful in designing and prioritising cost-effective interventions. Future research would benefit from addressing a broader range of climate exposures and including diverse populations, particularly vulnerable and displaced groups.

### 4.2. Future Opportunities

This report identifies several limitations in the existing research on climate change adaptation that have implications for future policy and practice for funders, research institutions, and policymakers. Firstly, the relatively limited number of studies evaluating adaptation interventions can be attributed to many projects primarily focusing on analysing descriptive data or performing impact assessments. Only a fraction (n = 17) of the articles reported conducting primary interventional research, where interventions were designed, implemented, and evaluated. In most cases, the projects primarily assessed the impact of climate change on health outcomes, with interventional research as a minor component. Funding calls should consider prioritising primary interventional research as a core component of future projects. The field should explore combined adaptation–mitigation interventions, with a move towards large-scale intervention studies with robust methodologies. Importantly, vulnerable groups, such as pregnant women and children, should be prioritised in such studies.

Despite the limitations, there were several examples of promising packages that could easily be scaled and implemented within national programmes: the Early-Warning Systems in the CCEHN, HEAT-SHIELD, and ClimApp projects; the provision of effective cooling interventions to vulnerable populations in the EXHAUSTION and PREP projects; and the behavioural changes and low-cost interventions protecting pregnant women in the CHAMNHA project, among others. Effectively and efficiently translating research into action will be a critical component of future climate and health action.

Overlapping thematic areas between the projects show the potential for improved collaboration. For instance, despite several of the Belmont and Horizons projects including a focus on Early Warning Systems in their study aims, the ClimApp project, which had developed such an intervention, was not utilised in other projects. We did not observe cross-references to other tools or experiences across the projects. This might have been due to the projects being implemented in parallel and not in sequence. Research institutions and funders should prioritise collaboration through multi-disciplinary consortiums.

The reviewed articles often did not address the pathways or mechanisms through which interventions operate. There was a stronger focus on interventions and outcomes, with less attention given to how interventions can be co-produced and implemented in collaboration with relevant stakeholders. With the urgent need to upscale climate change adaptation, understanding the most effective means to implement broad and diverse interventions can be just as important as the interventions themselves and conducted through process evaluations. Although few publications reported on beneficial interactions among policymakers, researchers, and the public, several studies utilised similar social science techniques which prioritised community engagement and co-design activities, which were central to intervention design and delivery. This presents an opportunity to ensure the suitability and applicability of interventions and challenges by increasing the time to design and deliver an intervention. Optimising these trade-offs will be a priority in future studies, but the approach in the included studies provides a roadmap to achieve this. Future research efforts should explore the roles and interactions between different stakeholders and sectors, investigating strategies to facilitate action at various levels and foster cooperation across multiple societal actors in reducing the impacts of climate change on health. Furthermore, fostering improved inter-regional collaboration will be critical in future research and any climate and health response.

### 4.3. Role of the Health Sector

Lastly, the results highlight the crucial role of the health sector in the net-zero transition and enhancing overall societal resilience. The health sector can contribute to carbon emission reduction, support adaptation efforts, and raise public awareness about climate change mitigation and adaptation [48,49]. Focusing on the health co-benefits and trade-offs of mitigation and adaptation policies is necessary to achieve this. Many climate mitigation and adaptation interventions positively affect human health (e.g., plant-based diets, active commuting, nature-based solutions), and future research should aim to understand these interactions better. Additionally, identifying supporting policies, as well as barriers and opportunities for implementation, is essential. With more robust evidence and supportive policies, the health sector can play an increasingly significant role in the transition to net-zero societies in the years to come.

### 4.4. Limitations

This review only examines the work conducted within the EU and the Belmont Forum. It may, therefore, be limited in its applicability to the entire field where other funding groups contribute research. Secondly, the projects reviewed were at different stages of the project lifecycle, making direct comparisons across projects challenging. Many projects were still in the final stages of data collection and analysis, and it is expected that additional publications may extend the research outputs of these projects. Delays caused by the COVID-19 pandemic further impacted the projects’ timelines.

## 5. Conclusions

This review provides a broad understanding of the scope and scale of research on adaptation interventions for climate change and health. While this review identifies critical shortcomings, it highlights opportunities for further research in key populations across under-represented regions. The benefits of collaboration and cross-cutting within a growing and evolving field cannot be overstated, with collaborative networks providing an important platform to coordinate research and pursue common priorities. By addressing these gaps, fostering interdisciplinary collaborations, and involving diverse stakeholders, we can develop a more comprehensive understanding of effective adaptation strategies and contribute to developing evidence-based policies and practices that equitably enhance human health resilience in the face of climate change. This review serves as an important platform to drive the scaled-up integration of effective adaptation interventions into national climate–health programmes.

## Figures and Tables

**Figure 1 ijerph-21-01565-f001:**
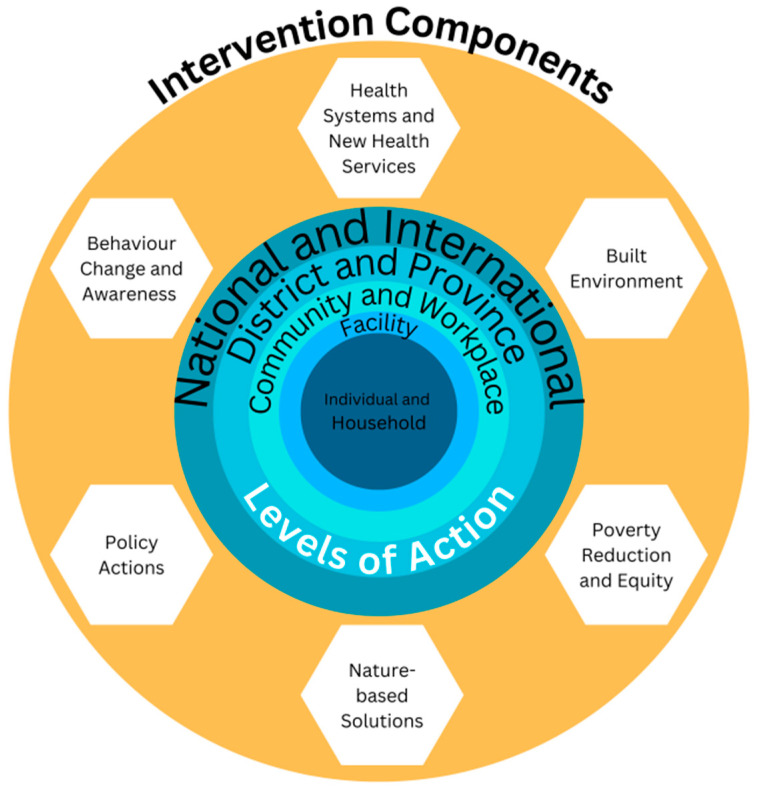
Framework for classifying interventions.

**Figure 2 ijerph-21-01565-f002:**
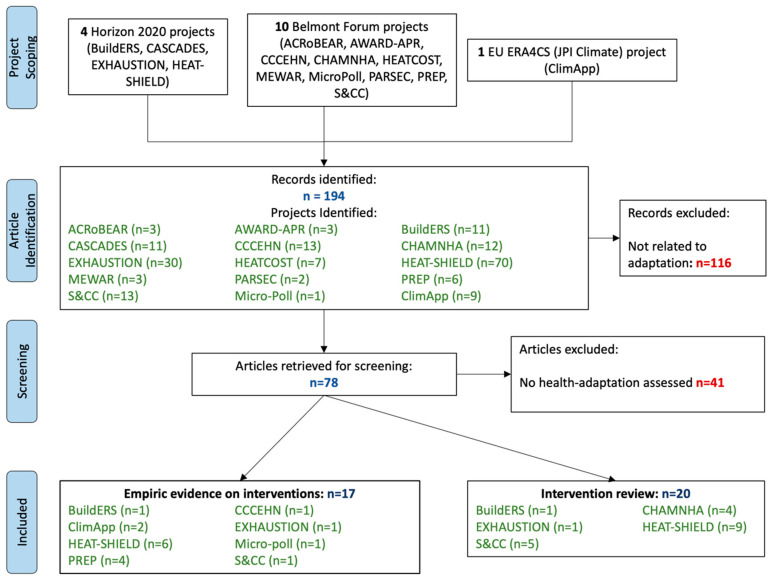
PRISMA flowchart.

**Figure 3 ijerph-21-01565-f003:**
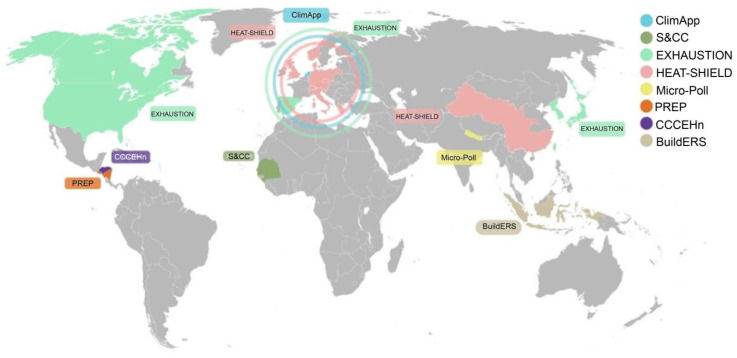
Map of distribution of projects by country.

**Figure 4 ijerph-21-01565-f004:**
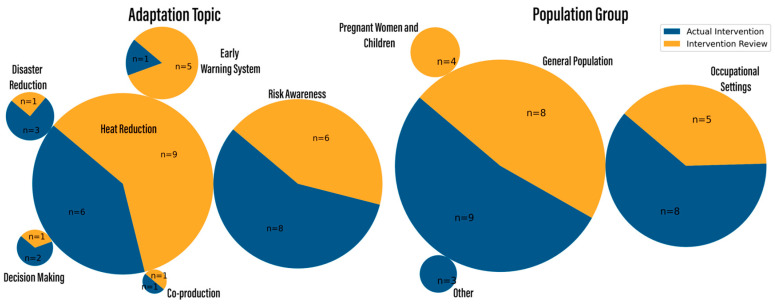
Distribution of study by adaptation topic and population group.

**Table 1 ijerph-21-01565-t001:** Summary of adaptations studied by the projects.

Project Name	Aims	Adaption Studied
BuildERS [9,10]	To strengthen the social capital, risk awareness, and preparedness of vulnerable segments of society.	Application of machine learning and data science methods to shorten response times, optimise resource allocation, and improve the risk awareness and resilience of vulnerable populations during disasters.
CCCEHN [11]	To identify the cascading risks of climate change in countries outside of Europe by supporting the design of a coherent European policy framework to address these risks.	Local systems developed by the community in complementing state systems and fostering long-term engagement and preparedness by means of risk awareness, collaborating with communities and local stakeholders, coproducing alert dissemination protocols, and developing plans to enhance existing response and contingency planning.
CHAMNHA [12,13,14,15]	To reduce the impacts of heat on birth and neonatal outcomes via intervention studies among pregnant women specifically aimed at reducing heat exposure in outdoor work during late pregnancy.	Information campaigns and behaviour change interventions that target pregnant women, female family members, community leaders, and other stakeholders to support self-care interventions during pregnancy. Investments in Africa in infrastructure, services, and human resources for maternal health for interventions such as Early Warning Systems.
ClimApp [16,17]	To combine weather forecast information with end-user data to provide heat and cold stress warnings. To facilitate the development of a decision support system to provide timely relevant guidelines to improve thermal resilience when adverse environmental conditions are expected.	The project assessed the usability of an app and conducted field validation studies, targeting specific groups such as the elderly and young children. This information can be used by individuals as guidance on how best to adapt their behaviour during periods of high temperatures to reduce any potential health risks associated with exposure.
EXHAUSTION [18,19]	To establish exposure-response functions for cardiopulmonary health outcomes of extreme temperatures, assess potential interactive effects of air pollutants, and project the future corresponding health burden for Europe.	Adaptation measures to reduce exposure to air pollution and heat stress, focusing on air conditioning as an effective heat adaptation strategy. The project also developed models for the formation and dispersion of plumes from major fires and evaluated approaches for monitoring heat-related deaths during heatwaves.
HEAT-SHIELD [20,21,22,23,24,25,26,27,28,29,30,31,32,33,34,35]	To develop correction equations to quantify outdoor work capacity and evaluate the performance implications of face mask use in hot environments.	The study quantified the impact of heat on human physical work capacity through existing heat indices and assessed the effects of prolonged face mask use in the heat on dyspnoea and motor cognitive performance.
Micro-Poll [46]	To quantify the nutritional value of an ecosystem and provide a framework to predict the effects of environmental change on human nutrition.	This project focused on food security and smallholder farms, assessing the nutritional value of ecosystems and the effects of environmental change on human nutrition.
PREP [42,43,44,45]	To assess preventive care and awareness around Chronic Kidney Disease of Undetermined Cause in outdoor workers who are highly vulnerable because of exposure to heat and dehydration.	The study reviews Adelante Initiative implementation outcomes, which comprise a package of services including water supply supplementation, rest in the shade, and improved ergonomics in the work settings of sugarcane workers, and evaluates their effectiveness in improving water consumption, productivity, and worker well-being.
S&CC [36,37,38,39,40,41]	To investigate the potential for aquaculture to reduce poverty and control schistosomiasis in Côte d’Ivoire (Ivory Coast) during an era of climate change.	The project evaluated the control of schistosomiasis through freshwater prawn aquaculture, removing vegetation which aided vector distribution, using artificial intelligence techniques to identify snails and parasites, community awareness and intervention campaigns, and co-benefit solutions such as altering land-use and irrigation practices.

## Data Availability

This research was conducted on publicly available information; however, full extraction tables are available upon request.

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
