# Peer review of "Scoping Review of Climate Change Adaptation Interventions for Health: Implications for Policy and Practice"

_ijerph, 2024, doi:10.3390/ijerph21121565_

Round 1
Reviewer 1 Report
Comments and Suggestions for Authors
Reviewer Report for Manuscript titled " Scoping Review of Climate Change Adaptation Interventions for Health: Implications for Policy and Practice
Minor revision is recommended to this submission. Specific comments are given below.
· Attempt seems to be good. The manuscript is generally well-structured, with clear sections outlining the research objectives, methods, results, and conclusions. However, some sections could benefit from improved clarity and organization.
· Lines 20-23 of the abstract highlight target populations and treatments, although there is a lack of attention on high-risk groups such as children and migrants. This limitation should be emphasized earlier in the abstract to align with the key findings in the results.
· The introduction gives a solid basis. However, the discussion of a lack of investment in health-related adaptation measures (Lines 54-56) might be improved with precise numbers on health financing gaps, followed by a more direct link to the study's objectives (Line 69).
· Although the application of PRISMA-ScR principles is noteworthy, the methodologies for data extraction and thematic analysis (Lines 86-94) are not completely defined. Consider describing the specific themes or categories utilized for classification, and any potential limits in the search method, especially given that only papers relevant to Belmont Forum and EU Horizon programs were included.
· Line 79 indicates a request for "missing information" from principal investigators. The process for dealing with incomplete data has to be clarified.
· The section offers complete statistics. However, using "n=15" in some research lacks depth. For example, techniques are briefly described in lines 126–132. More information is needed on how mixed techniques and cost-effectiveness studies influenced the outcomes.
· Line 135 highlights the broader populace as the main emphasis. More detailed details about sub-groups in the results would be helpful in aligning with the introduction's request for high-risk populations.
· Lines 122-125 provide insight into geographic distribution, but future studies should address the under-representation of the global south.
· The manuscript references to Figures 1 and 4 (Lines 97, 161), however further explanation is needed in the text to help readers understand the findings.
· Line 164 indicates that further figures are available in Supplementary File 2. To improve clarity, a brief description of these graphics may be included in the main paper.
· This part provides a balanced discussion, but it lacks a clear conclusion on the practical consequences for policy. For example, the discussion of interventions such as early warning systems (Lines 245-246) could be developed to illustrate how these could be scaled at the national or worldwide levels.
· The limitations (Lines 309-316) are acknowledged, however future studies should address these problems, including restricted inter-regional collaboration.
· The conclusion summarises the findings well, however explicit policy recommendations would be beneficial. For example, tying the HEAT-SHIELD project's findings (Lines 192-204) to global public health policies would make the conclusion more actionable.
· The manuscript references are appropriate and up to date. However, in-text citations are not always completely clarified, particularly when alluding to policy consequences (lines 270-272). It would be beneficial to link these citations to specific policy frameworks in order to increase the manuscript's relevance to practitioners.
· References should be arranged as per the journal format.
· These corrections would improve the content of the manuscript and significantly increase the citations
Minor editing of English language required.
Author Response
Please see attached responses to all reviewers.

Reviewer 2 Report
Comments and Suggestions for Authors
I congratulate the authors for comming out with out of the Box thought and very impressive MS on efforts taken care by Scientntific community on interventions in climate change adaptations on health. This particular research cum review gives insights for policy makers as well as scinetific community for for making effoert in policy making and research for targeting a particular group. as it show common population is targated very leas only specific grroups like pregnanat women and other group are targeted in till date research but in broad spectrum climate change severly impact at large scale irrespective of a particular group. so need to be targated whole population.
Comments on the Quality of English LanguageThe english langage is fairly good and well written MS
Author Response

(The authors gave the same response as above.)

Reviewer 3 Report
Comments and Suggestions for Authors
The authors have organized the paper well. Much statistical analysis is not observed in the study. It would be much interesting if some advanced statistical analysis can be done from this study.
Author Response

(The authors gave the same response as above.)

Reviewer 4 Report
Comments and Suggestions for Authors
The authors used Chat GPT for initial report creation. Chat GPT usually produce fake citation. Did the authors verified the citation.
The authors grab information from literature. Their own contribution is missing. Update the paper to make it more readable and informative.
The authors should highlight spatial hotspot particularly emphasizing on urban and rural area population.
The authors highlighted the importance of nature-based solutions in making the community climate resilient. Kindly mention the specific nature based solutions which helps to make the community climate resilient.
Comments on the Quality of English Languageminor editing
Author Response

(The authors gave the same response as above.)

Reviewer 5 Report
Comments and Suggestions for Authors
1) The primary motivation of the study was not clear.
2) The authors should uncover why there is a need to conduct this study in the introduction.
3) Avoid using “we, we, our etc.”
4) There are several grammatical errors and spelling errors in the study.
5) The conclusion section of the study lacks sound policy proposals.
Comments on the Quality of English LanguageMinor editing of English language required.
Author Response

(The authors gave the same response as above.)
